# Local Variation Matters: A Diagnostic Evaluation of Time Series Forecast Metrics

## Abstract

MSE and MAE remain useful default measures for time series forecasting, but pointwise accuracy does not say whether a forecast keeps the temporal structure of the target. A low-error forecast can still flatten local fluctuations, miss peaks and troughs, or shift abrupt changes. Prior work has used Dynamic Time Warping (DTW) and the Temporal Distortion Index (TDI) to expose shape and timing effects that are hidden by MSE. We study a complementary issue: loss of local variation. We evaluate DLinear, PatchTST, TimeMixer, iTransformer, Chronos-2, and TimesFM on univariate target-only forecasting tasks from ETT, Weather, and ESNet telemetry. Alongside MSE, MAE, FastDTW, and TDI, we introduce two ratios: Total Variation Ratio (TVR), which compares accumulated local movement, and High-Frequency Energy Ratio (HFER), which compares residual energy after a moving-average trend is removed. We use these ratios as checks on smoothing, not as a replacement for existing metrics. The results show that strong pointwise scores can coincide with weak TVR and HFER, especially for pretrained foundation-model forecasts. Simpler baselines sometimes retain more local movement even when their pointwise errors are larger. On ESNet, all evaluated models miss much of the spike-like behavior, indicating that heavy-tailed telemetry also needs application-specific event or tail metrics. The study supports analyzing pointwise error, temporal alignment, and local-variation measures together when forecast shape matters.

## 1 Introduction

Time series forecasting papers commonly report mean squared error (MSE) and mean absolute error (MAE). These metrics are useful: they are simple, stable under repeated evaluation, and easy to compare across models. They also match the standard supervised learning setup, where a model receives a fixed history window and is asked to predict the next set of values. For many applications, this pointwise view is appropriate. If the main concern is the numerical value at each timestamp, then MSE and MAE provide direct summaries of forecast error.

The difficulty is that a forecast is often used as a trajectory, not only as a collection of independent point estimates. A prediction can have a reasonable average error while still changing the temporal behavior of the signal. Peaks may be damped, troughs may be lifted, abrupt changes may be delayed, and local fluctuations may be replaced by a smoother curve. These differences are not minor visual details when the downstream task depends on the shape of the series. They affect whether an analyst sees a burst, a transition, a period of instability, or a change in operating regime.

This issue appears in several scientific and operational settings. In particle-physics experiments such as NOvA (Adamson et al., 2016), a galactic core-collapse supernova would appear as a short burst of candidate events rather than as a slowly varying trend. A forecast or detector-side prediction that smooths such a burst could obscure the timing and duration of the event. In scientific network telemetry, sudden changes in bandwidth can reflect changes in data movement behavior, congestion, or application activity. In financial forecasting, the timing and persistence of volatility bursts can matter even when the average forecast error is small. The ESNet telemetry used in this paper (Carder et al., 2022) is not detector-event data, and

we do not use it as a proxy for NOvA physics signals. We use it as a scientific-network case study with intermittent, high-dynamic-range behavior, where the limitations of aggregate pointwise error are easy to see. Prior studies (Chari et al., 2025) have shown that most state of the art time series forecasting and imputation models do not accurately capture the bursts in this kind of data, so models that are built to optimize metrics that highlight the importance of the bursts might perform better.

There is already substantial evidence that MSE alone is not sufficient for forecast evaluation. CONTIME (Jhin et al., 2024), for example, studies prediction delay using MSE together with Dynamic Time Warping (DTW) and the Temporal Distortion Index (TDI). DTW measures how well two sequences can be matched when local time warping is allowed. TDI measures how far the resulting alignment moves away from matching equal timestamps. Together, these metrics separate pointwise error from temporal alignment and timing displacement. This is an important evaluation perspective because two forecasts with similar MSE can have different delays, and a forecast can match a shape only after being warped in time.

This paper focuses on a related but separate issue: over-smoothing. A forecast can be well aligned in time and still contain too little local movement. It can follow the broad level of the target while suppressing the variation that makes the series locally informative. Conversely, a forecast can preserve more local movement while having worse pointwise error because its level is biased or because some structures are shifted. Temporal alignment and local-variation preservation therefore describe different aspects of the same forecast. Both are useful, but neither one subsumes the other.

We make this distinction quantitative using two simple ratios. Total Variation Ratio (TVR) compares the discrete total variation of the forecast with that of the target, where total variation is the sum of absolute first differences (Rudin et al., 1992). If TVR is far below one, the forecast changes less from step to step than the target. High-Frequency Energy Ratio (HFER) first removes a centered moving-average trend and then compares the standard deviation of the remaining residuals (Carbone et al., 2004). If HFER is far below one, the forecast has less short-scale residual energy than the target. We use these ratios as additional evidence about forecast behavior. They are not meant to replace MSE, MAE, DTW, or TDI, and a value near one is not by itself a claim that the forecast is accurate.

The goal of the paper is not to identify a single best model under a new metric. Instead, we ask how conclusions change when the same forecasts are read through pointwise, alignment-aware, and local-variation metrics. We evaluate DLinear (Zeng et al., 2022), PatchTST (Nie et al., 2023), TimeMixer (Wang et al., 2024), iTransformer (Liu et al., 2024), Chronos-2 (Ansari et al., 2025), and TimesFM (Das et al., 2024). The experiments use univariate target-only forecasting on the ETT benchmarks (Zhou et al., 2021), Weather (Kolle), and ESNet telemetry (Carder et al., 2022). This setup keeps the metric comparison direct because each model receives the same target history and forecasts the same target variable. It also limits the interpretation of models such as iTransformer, which are designed primarily for multivariate forecasting.

The paper makes four contributions. First, it separates pointwise error, temporal alignment, and local-variation preservation as distinct aspects of forecast behavior. Second, it defines TVR and HFER as lightweight checks for whether local movement has been damped or exaggerated. Third, it evaluates these metrics across six forecasting models, standard benchmark datasets, and a scientific-network telemetry case study. Fourth, it shows that the apparent strength of a model can change with the metric being emphasized, especially when the target contains local fluctuations, abrupt changes, or heavy-tailed spikes.

## 2 Background

### 2.1 Error metrics for time series forecasting

Most time series forecasting studies report pointwise errors such as mean squared error (MSE) and mean absolute error (MAE). Let $y = (y_1, \ldots, y_T)$ denote the ground-truth sequence over a prediction horizon of length $T$, and let $\hat{y} = (\hat{y}_1, \ldots, \hat{y}_T)$ denote the corresponding forecast. The two standard pointwise metrics are

$$\text{MSE}(y, \hat{y}) = \frac{1}{T} \sum_{t=1}^{T} (y_t - \hat{y}_t)^2, \tag{1}$$

and

$$\mathrm{MAE}(y, \hat{y}) = \frac{1}{T} \sum_{t=1}^{T} |y_t - \hat{y}_t|. \tag{2}$$

These metrics compare the prediction and ground truth at the same timestamp. MSE gives larger weight to large errors because the residual is squared, while MAE grows linearly with the absolute residual. Both are useful measures of pointwise accuracy, but neither metric distinguishes between an error caused by incorrect amplitude and an error caused by a small temporal shift.

To measure shape similarity under temporal shifts, we also consider Dynamic Time Warping (DTW). Define the pairwise cost matrix $C \in \mathbb{R}^{T \times T}$ by

$$C_{i,j} = d(y_i, \hat{y}_j), \tag{3}$$

where $d(\cdot, \cdot)$ is a local discrepancy, which we take to be the absolute difference or squared difference depending on the implementation. A warping path is a sequence $\pi = ((i_1, j_1), \dots, (i_L, j_L))$ satisfying boundary, monotonicity, and step-size constraints:

$$(i_1, j_1) = (1, 1), \qquad (i_L, j_L) = (T, T), \tag{4}$$

and

$$(i_{\ell+1} - i_\ell, \; j_{\ell+1} - j_\ell) \in \{(1, 0), (0, 1), (1, 1)\} \tag{5}$$

for $\ell = 1, \dots, L - 1$. Let $\mathcal{A}(T, T)$ denote the set of all valid warping paths between two length-$T$ sequences. The DTW distance is the minimum accumulated alignment cost:

$$\mathrm{DTW}(y, \hat{y}) = \min_{\pi \in \mathcal{A}(T,T)} \sum_{(i,j) \in \pi} C_{i,j}. \tag{6}$$

Equivalently, it can be computed by dynamic programming. If $D_{i,j}$ is the optimal cost for aligning prefixes $(y_1, \dots, y_i)$ and $(\hat{y}_1, \dots, \hat{y}_j)$, then

$$D_{i,j} = C_{i,j} + \min \{D_{i-1,j}, D_{i,j-1}, D_{i-1,j-1}\}, \tag{7}$$

with the usual boundary initialization. The DTW value is $D_{T,T}$, possibly normalized by the path length or horizon length. A low DTW score indicates that the two sequences can be matched with small cumulative cost after allowing local stretching or compression of the time axis.

DTW measures the cost of the best alignment, but it does not by itself indicate how far that alignment moves away from matching equal timestamps. For this reason, we also use the Temporal Distortion Index (TDI), following the idea of measuring the temporal displacement induced by the DTW alignment. Let $\pi^\star$ be an optimal DTW path:

$$\pi^\star \in \arg \min_{\pi \in \mathcal{A}(T,T)} \sum_{(i,j) \in \pi} C_{i,j}. \tag{8}$$

The TDI is then defined as the average squared deviation of the alignment path from the diagonal:

$$\mathrm{TDI}(y, \hat{y}) = \frac{1}{|\pi^\star| T^2} \sum_{(i,j) \in \pi^\star} (i - j)^2. \tag{9}$$

The factor $T^2$ makes the score comparable across horizons, while $|\pi^\star|$ averages over the number of matched pairs in the path. A forecast whose events occur at the correct times will have an alignment path near the diagonal and therefore a small TDI. A forecast that has a similar shape but is delayed or advanced in time can have a larger TDI even when its DTW score is relatively small.

In addition to pointwise error and temporal alignment, we measure whether a forecast preserves the local variation of the target sequence. We first use Total Variation Ratio (TVR), which we define from the discrete total variation of a sequence. For a sequence $x = (x_1, \dots, x_T)$, let

$$\mathrm{TV}(x) = \sum_{t=2}^{T} |x_t - x_{t-1}|. \tag{10}$$

This is the one-dimensional discrete form of total variation, which has been widely used as a measure of accumulated local change in signal and image processing (Rudin et al., 1992). We define TVR as

$$\text{TVR}(y, \hat{y}) = \frac{\text{TV}(\hat{y})}{\text{TV}(y) + \epsilon}, \tag{11}$$

where $\epsilon > 0$ is a small constant used only to avoid division by zero. A TVR below one indicates that the forecast has less total local movement than the target, while a TVR near one indicates a similar amount of local movement. A TVR above one indicates that the forecast is more locally variable than the target.

We also define High-Frequency Energy Ratio (HFER) to compare short-scale fluctuation energy after removing a local trend. Let $k$ be an odd moving average window size. For each timestamp $t$, define the local window

$$W_t = \left\{ s \in \{1, \dots, T\} : |s - t| \leq \frac{k-1}{2} \right\}. \tag{12}$$

The centered moving average of a sequence $x$ is then

$$\text{MA}_k(x)_t = \frac{1}{|W_t|} \sum_{s \in W_t} x_s. \tag{13}$$

The local residual after removing this moving-average trend is

$$r_t(x; k) = x_t - \text{MA}_k(x)_t. \tag{14}$$

Using these residuals, we define HFER as

$$\text{HFER}(y, \hat{y}; k) = \frac{\text{std}\left(r(\hat{y}; k)\right)}{\text{std}\left(r(y; k)\right) + \epsilon}. \tag{15}$$

This follows the common use of moving-average detrending to separate local fluctuations from slower trend components (Carbone et al., 2004). A HFER below one indicates that the forecast has less short-scale fluctuation energy than the target, while a value near one indicates similar residual energy after trend removal. A HFER above one indicates that the forecast contains more short-scale residual variation than the target.

The six metrics therefore emphasize different aspects of forecast quality. MSE and MAE measure pointwise error at fixed timestamps. DTW measures how well the forecast can be aligned to the target sequence when local time shifts are allowed. TDI measures the amount of temporal displacement required by that alignment. TVR measures whether the total amount of local movement is damped or exaggerated. HFER measures whether short-scale residual fluctuation energy is damped or exaggerated after subtracting a moving-average trend. In rapidly varying series, these distinctions are important: a smoothed forecast can have good pointwise error while suppressing peaks and troughs, whereas a forecast with more local movement can preserve short-scale structure but still have worse pointwise error.

A practical limitation of standard DTW is its quadratic time and memory cost in the sequence length. This can be burdensome for long horizons or for large benchmark suites. Several alternatives have been proposed to reduce this cost. FastDTW (Salvador & Chan, 2007) approximates DTW through a multiresolution procedure, although later work cautions that it is not always faster or preferable to optimized exact DTW in realistic settings (Wu & Keogh, 2021). Other variants, such as Segmental DTW (Tsai, 2021) and Tralie & Dempsey (2020), exploit alternative decompositions or parallel computation to accelerate sequence alignment. These methods are relevant options when alignment metrics must be computed at larger scale. In this study, we use FastDTW because even though it is slower than standard DTW in edge cases, it is faster on average and has a well-supported Python package.

## 2.2 Forecasting models considered

We include six models that represent common choices in current forecasting work: a linear decomposition baseline, transformer variants, an MLP-based multiscale model, and two pretrained time series foundation models.

**DLinear.**   DLinear (Zeng et al., 2022) comes from the LTSF-Linear family. It first separates the input into trend and residual components, then applies linear mappings to produce the forecast. We include it because it is a strong and inexpensive baseline, and because its behavior is often easier to interpret than that of larger neural models.

**PatchTST.**   PatchTST (Nie et al., 2023) applies a transformer to patches of each univariate series rather than to individual time points. The patching step reduces the effective sequence length seen by attention and allows the model to use longer histories. In the channel-independent version used in many benchmarks, the same model is shared across variables; in our experiments the input is only the target series.

**TimeMixer.**   TimeMixer (Wang et al., 2024) is an MLP-based forecasting model that mixes information across multiple temporal scales. Its design separates past representations into components at different scales and combines predictors for the future horizon. We include it as a recent non-attention architecture that is intended to capture both short- and longer-scale patterns.

**iTransformer.**   iTransformer (Liu et al., 2024) changes the usual transformer tokenization for multivariate time series by treating variables as tokens and using the historical segment as the token representation. This design is meant to exploit cross-variable relationships. Since our study uses target-only forecasting, the iTransformer results should be read as a controlled univariate comparison rather than as a full test of the architecture.

**Chronos-2.**   Chronos-2 (Ansari et al., 2025) is a pretrained time series foundation model that produces probabilistic forecasts in a zero-shot setting. For the deterministic metrics in this paper, we use its median forecast. Its quantile forecasts are used only for the plots.

**TimesFM.**   TimesFM (Das et al., 2024) is a pretrained decoder-only foundation model for time series forecasting. It provides a second pretrained baseline, allowing us to compare foundation-model forecasts with models trained directly on the benchmark data.

This model set is not intended to exhaust the forecasting literature. It is a representative set for asking whether different metric families lead to the same interpretation of forecast behavior.

## 2.3   CONTIME and evaluation with DTW and TDI

CONTIME (Jhin et al., 2024) studies prediction delay in time series forecasting. Its evaluation combines MSE with DTW and TDI, so that pointwise accuracy, shape matching, and temporal displacement are reported together. The results show that models with similar MSE can differ substantially in their timing behavior. This paper follows the same general motivation for using more than one metric, but it focuses on smoothing rather than delay. DTW and TDI describe the alignment path between two sequences; TVR and HFER instead summarize how much local movement remains in the forecast.

# 3   Experimental Setup

We evaluate all models on univariate target-only forecasting. Each model is given the past values of one target variable and predicts future values of that same variable. Table 1 lists the datasets and splits.

For ETTh1, ETTm1, and ETTm2, the target is oil temperature (OT). For Weather, the target column is named `OT` in the LTSF-formatted benchmark file (Kwangryeol Park, 2024). The source measurements come from the Max Planck Institute for Biogeochemistry weather station in Jena and include 10-minute observations of air temperature, humidity, pressure, wind, radiation, and precipitation (Kolle). Since the benchmark file does not expand the abbreviation `OT`, we refer to the target by its column name. For ESNet, the target is outbound bandwidth computed from the `ifHCOutOctets` counter and expressed in bits per second.

Table 1: Dataset metadata used in the experiments. The variable count excludes the timestamp column for ETT and Weather.

| Dataset | Target | Frequency | Input mode | Train | Validation | Test |
|---------|--------|-----------|------------|-------|------------|------|
| ETTh1 | OT | 1 hour | univariate target-only | 8640 | 2880 | 2880 |
| ETTm1 | OT | 15 minutes | univariate target-only | 34560 | 11520 | 11520 |
| ETTm2 | OT | 15 minutes | univariate target-only | 34560 | 11520 | 11520 |
| Weather | `OT` target column | 10 minutes | univariate target-only | 36792 | 5271 | 10540 |
| ESNet | `ifHCOutOctets` to bits/s | 5 minutes | univariate target-only | 51808 | 12953 | 4321 |

For the benchmark datasets, the lookback length is 336 and the prediction lengths are 96, 336, and 720. We do not use the common 192-step setting because our goal is to compare metric behavior at short, medium, and long horizons. For ESNet, the forecast horizon is 4321 samples, corresponding to 15 days on the processed 5-minute grid.

DLinear, PatchTST, TimeMixer, and iTransformer are trained through the NeuralForecast implementation where applicable (Olivares et al., 2022). Chronos-2 and TimesFM are evaluated zero-shot. TimesFM is included for ETT and Weather, but not for ESNet, because the 4321-step ESNet horizon exceeds the 1024-step horizon used by the implementation in our notebook.

All reported scores are computed on scaled target values. The scaler is fit on the training split and then applied to validation and test windows. Because we score only the target column, the MSE and MAE values in this paper are not meant to reproduce the multivariate aggregate scores reported in the original model papers. The within-paper comparison is consistent because all models use the same target, origins, scaling, and metric code.

For each dataset, model, and horizon, we compute MSE, MAE, FastDTW, TDI, TVR, and HFER. FastDTW is normalized by alignment path length, and TDI is computed from the same path. TVR compares total variation in the forecast and target; HFER compares residual standard deviation after subtracting a centered moving-average trend. We evaluate up to six non-overlapping test windows for the benchmark datasets and three 15-day windows for ESNet. Rolling-origin averages are used for model-level comparisons, while the single-horizon tables explain the plotted examples.

Our code is publicly available at [1].

## 4 Results

For MSE, MAE, DTW, and TDI, smaller values are preferred. TVR and HFER are read differently: values near one mean that the forecast contains a similar amount of local movement or residual fluctuation energy as the target. Values below one indicate smoothing, and values above one indicate excess local variation.

We report both single-horizon and rolling-origin results. The single-horizon values correspond to the figures and are used to explain the plotted forecasts. The rolling-origin values summarize behavior over six non-overlapping test windows and are the basis for model-level comparisons. The standard deviations are therefore variation over forecast origins, not uncertainty estimates.

The main text shows ETTh1 at horizon 96, ETTm1 at horizons 336 and 720, and ESNet at the 15-day horizon. These cases give one short benchmark horizon, two longer benchmark horizons, and a heavy-tailed telemetry example. The remaining benchmark settings are discussed in Appendix A.1.

### 4.1 Prediction length 96

Figure 1 shows one ETTh1 forecast at horizon 96. The example is useful because the pointwise and local-variation metrics tell different stories. TimesFM has the strongest pointwise scores on this horizon, with MSE

---

[1]https://anonymous.4open.science/r/Rethink_time_series_forecasting_metrics-262E

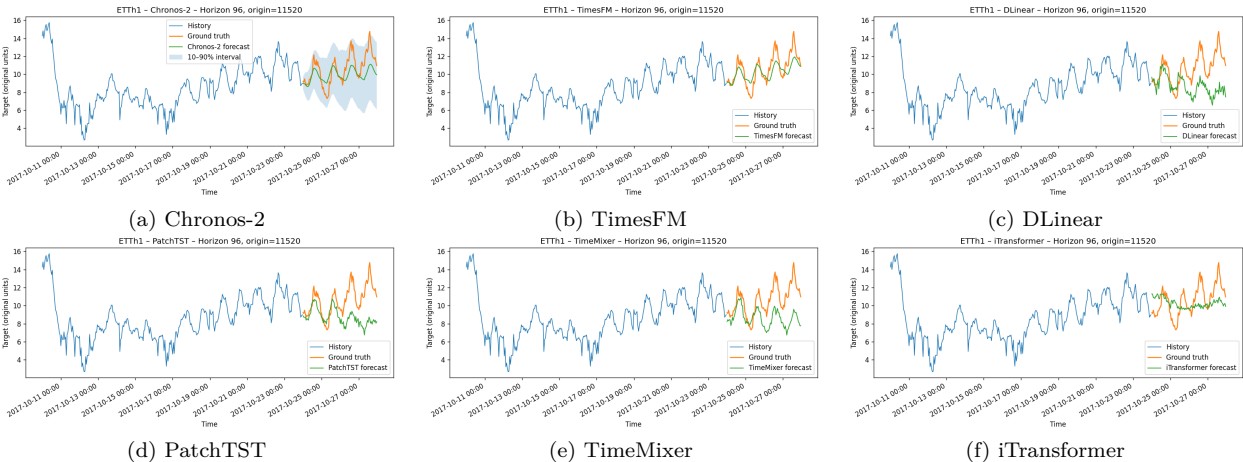

Figure 1: Forecasts on ETTh1 for prediction length 96 on a representative test horizon. TimesFM and Chronos-2 obtain strong pointwise scores, but their forecasts are smoother than the target. DLinear has worse pointwise error on this horizon, but its TVR and HFER are closer to one.

Table 2: ETTh1 results for prediction length 96 on the exact horizon shown in Figure 1. MSE, MAE, DTW, and TDI are lower-is-better. TVR and HFER are best when they are close to one.

| Model | MSE | MAE | DTW | TDI | TVR | HFER |
|---|---|---|---|---|---|---|
| Chronos-2 | 0.0214 | 0.1142 | 0.0720 | 0.0375 | 0.3153 | 0.4395 |
| TimesFM | **0.0139** | **0.0930** | **0.0445** | 0.0012 | 0.3463 | 0.4686 |
| DLinear | 0.0841 | 0.2354 | 0.1548 | 0.0020 | **1.0299** | **0.7268** |
| PatchTST | 0.0775 | 0.2186 | 0.1476 | **0.0011** | 0.6764 | 0.5701 |
| TimeMixer | 0.0726 | 0.2238 | 0.1447 | 0.0012 | 0.6400 | 0.6413 |
| iTransformer | 0.0356 | 0.1548 | 0.0964 | 0.0214 | 0.4544 | 0.3049 |

0.0139 and MAE 0.0930, and it also has the lowest DTW score, 0.0445. However, its TVR is 0.3463 and its HFER is 0.4686, which means that it preserves much less local movement and short-scale residual energy than the target. Chronos-2 shows similar behavior, with low pointwise error but low TVR and HFER.

DLinear has worse MSE, MAE, and DTW on this horizon, but it is closer to the target in local-variation terms. Its TVR is 1.0299, which is closest to one among the evaluated models, and its HFER is 0.7268, also the closest to one on this horizon. PatchTST obtains the lowest TDI, which means that its alignment path stays closest to the diagonal, but its TVR and HFER are still below one. Thus, a small timing distortion score can coexist with substantial damping of local amplitude.

The six-window rolling-origin results give a similar summary. iTransformer has the lowest average MSE, 0.0980, while TimesFM has the lowest average MAE and DTW, 0.2515 and 0.1739. PatchTST has the lowest average TDI, 0.0079. However, DLinear is closest to the target under the local-variation diagnostics, with average TVR 0.8221 and average HFER 0.6248. The pointwise and alignment metrics therefore do not identify the same behavior as TVR and HFER.

## 4.2 Prediction length 336

Figure 2 shows an ETTm1 forecast at horizon 336. Here the plotted behavior and the single-horizon metrics agree more closely. DLinear has the lowest MSE, MAE, DTW, and TDI on this horizon. It also has the TVR

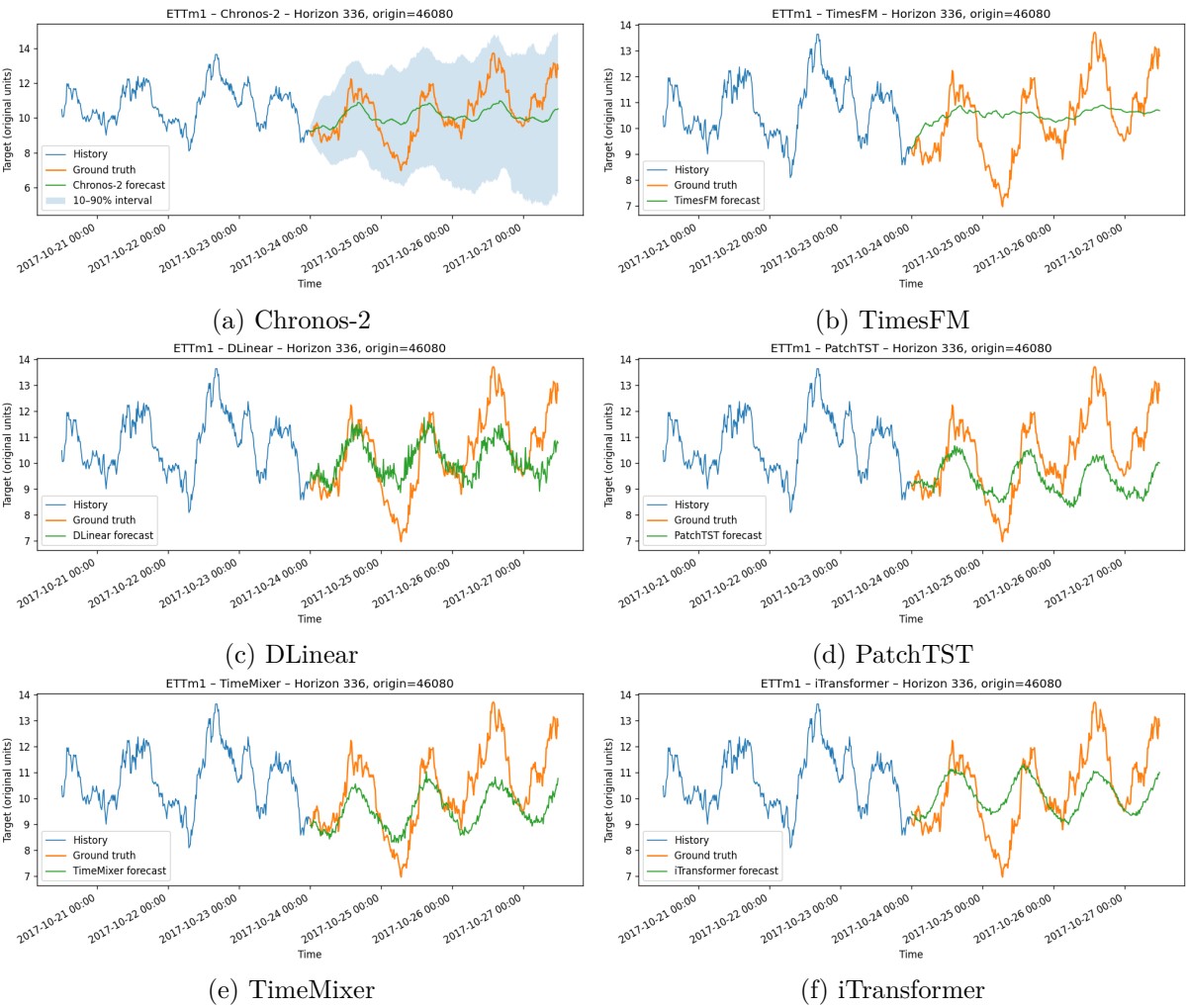

Figure 2: Forecasts on ETTm1 for prediction length 336 on a representative test horizon. DLinear follows the local up-and-down structure most closely on this horizon and is also strongest under the reported single-horizon metrics. Chronos-2 and TimesFM track the broad level but smooth much of the short-term variation.

closest to one and the HFER closest to one. Thus, for this plotted horizon, DLinear is not only better under pointwise and alignment metrics, but also better at preserving local variation.

Chronos-2 has the second-lowest MSE and MAE, but its forecast is smoother than the target. Its TVR is 0.1613 and its HFER is 0.3248. TimesFM is even smoother on this horizon, with TVR 0.0900 and HFER 0.1643. These ratios indicate that the foundation-model forecasts follow the broad level while damping much of the short-scale movement.

The rolling-origin results over six ETTm1 windows are less uniform than the single plotted horizon. Chronos-2 has the lowest average MSE and MAE, 0.1564 and 0.3071, while TimesFM has the lowest average DTW, 0.2123. TimeMixer has the lowest average TDI, 0.0076. DLinear remains closest under the local-variation diagnostics, with average TVR 1.1281 and average HFER 0.5016. This mismatch is the reason we use rolling-origin averages for model-level comparisons and the plotted horizons only for interpretation.

Table 3: ETTm1 results for prediction length 336 on the exact horizon shown in Figure 2.

| Model | MSE | MAE | DTW | TDI | TVR | HFER |
|---|---|---|---|---|---|---|
| Chronos-2 | 0.0183 | 0.1073 | 0.0555 | 0.0024 | 0.1613 | 0.3248 |
| TimesFM | 0.0226 | 0.1206 | 0.0731 | 0.0022 | 0.0900 | 0.1643 |
| DLinear | **0.0165** | **0.0993** | **0.0379** | **0.0015** | **1.2706** | **0.6199** |
| PatchTST | 0.0285 | 0.1387 | 0.0592 | 0.0023 | 0.5324 | 0.4720 |
| TimeMixer | 0.0205 | 0.1133 | 0.0410 | 0.0018 | 0.5397 | 0.4562 |
| iTransformer | 0.0183 | 0.1074 | 0.0437 | 0.0016 | 0.3156 | 0.3726 |

Table 4: ETTm1 results for prediction length 720 on the exact horizon shown in Figure 3.

| Model | MSE | MAE | DTW | TDI | TVR | HFER |
|---|---|---|---|---|---|---|
| Chronos-2 | 0.0346 | 0.1457 | 0.0827 | 0.0104 | 0.1646 | 0.2940 |
| TimesFM | 0.0374 | 0.1526 | 0.0971 | 0.0020 | 0.0674 | 0.1164 |
| DLinear | **0.0272** | **0.1317** | **0.0536** | 0.0058 | **1.2332** | **0.5228** |
| PatchTST | 0.0344 | 0.1535 | 0.0707 | 0.0771 | 0.6097 | 0.4975 |
| TimeMixer | 0.0325 | 0.1475 | 0.0629 | 0.0132 | 0.5725 | 0.4951 |
| iTransformer | 0.0299 | 0.1418 | 0.0610 | **0.0006** | 0.3182 | 0.4465 |

### 4.3 Prediction length 720

Figure 3 shows ETTm1 at horizon 720. At this longer horizon, the target contains larger peaks and drops. DLinear again has the lowest MSE, MAE, and DTW on the plotted horizon. It also has the TVR closest to one and the HFER closest to one. iTransformer has the lowest TDI, but its TVR is 0.3182, which means that it still suppresses much of the local variation. TimesFM has the smoothest forecast on this horizon, with TVR 0.0674 and HFER 0.1164.

The rolling-origin results over six ETTm1 windows point in the same direction. DLinear has the lowest average MSE and MAE, 0.1335 and 0.2833, and also the lowest average TDI, 0.0066. iTransformer has the lowest average DTW, 0.1684. DLinear remains closest under the local-variation diagnostics, with average TVR 1.2897 and average HFER 0.4939. TimesFM has the lowest average TVR, 0.0581, and one of the lowest HFER values, 0.1457, which is consistent with a much smoother forecast at this horizon.

### 4.4 ESNet telemetry

Figure 4 shows the 15-day ESNet forecast horizon. This setting is harder than the ETT examples. The target contains abrupt changes over several orders of magnitude. None of the models captures these changes reliably. TimesFM is not included for this dataset because the 4321-step ESNet horizon exceeds the 1024-step horizon limit used by the TimesFM implementation in our experiments.

The pointwise and alignment metrics still select different models. On the plotted horizon, iTransformer has the lowest MSE, while DLinear has the lowest MAE and DTW. PatchTST has the lowest TDI. These scores should not be read as evidence that the task is solved. The local-variation ratios make the failure easier to see. Even the highest TVR and HFER values are far below one. TimeMixer has the TVR closest to one on the plotted horizon, but the value is only 0.0430. It also has the HFER closest to one, but the value is only

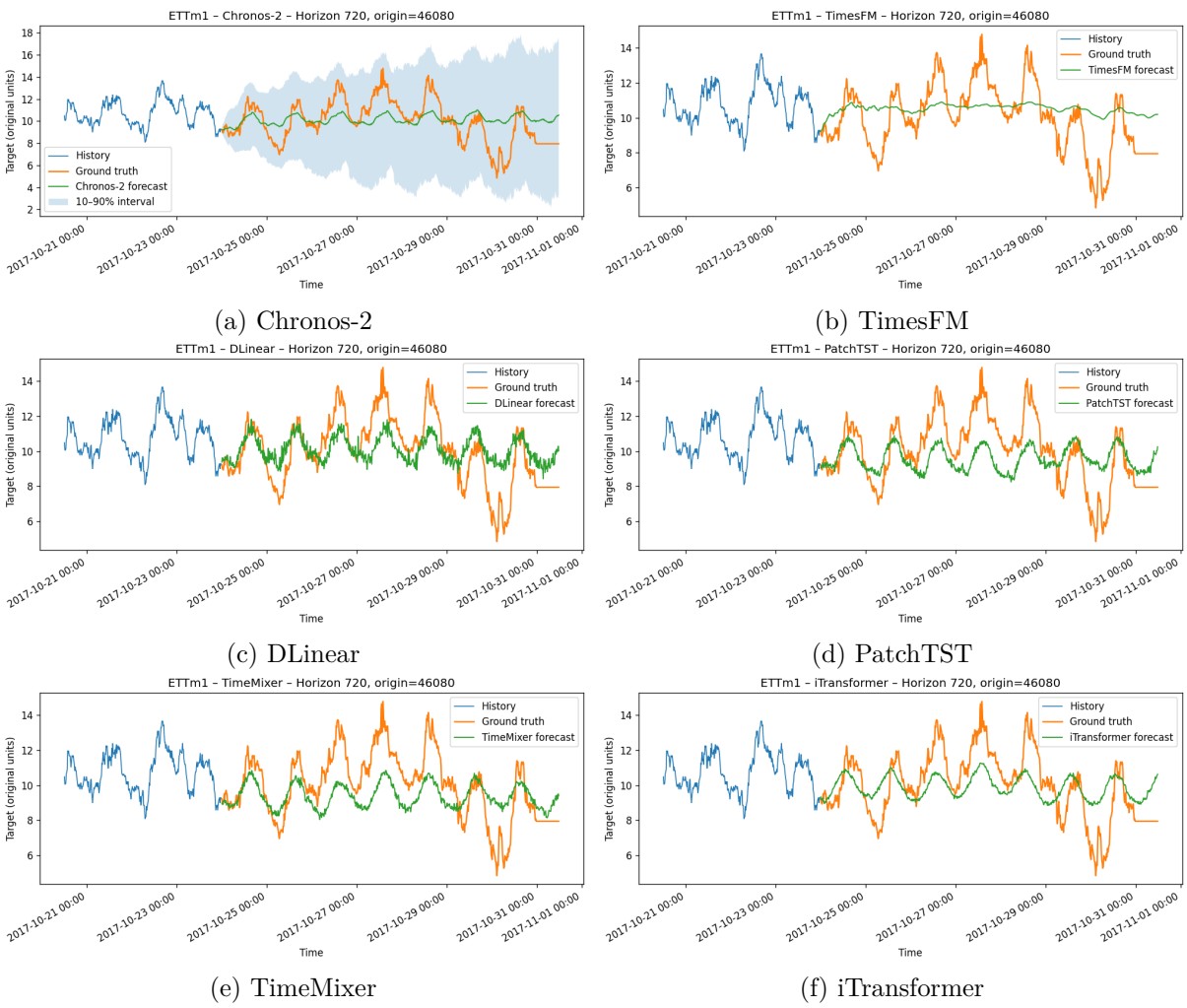

(a) Chronos-2         (b) TimesFM

(c) DLinear         (d) PatchTST

(e) TimeMixer         (f) iTransformer

Figure 3: Forecasts on ETTm1 for prediction length 720 on a representative test horizon. DLinear preserves more of the local up-and-down structure than the pretrained foundation models on this horizon. TimesFM and Chronos-2 remain smoother than the target.

Table 5: ESNet results for the exact 15-day horizon shown in Figure 4. TimesFM is omitted because the 4321-step horizon exceeds the TimesFM horizon limit used in our experiments.

| Model | MSE | MAE | DTW | TDI | TVR | HFER |
|---|---|---|---|---|---|---|
| Chronos-2 | 1.3222 | 0.1878 | 0.1649 | 0.0013 | 0.0013 | 0.0008 |
| DLinear | 1.3216 | **0.1863** | **0.1358** | 0.0028 | 0.0033 | 0.0009 |
| PatchTST | 1.3254 | 0.1974 | 0.1805 | $\mathbf{1.47{\times}10^{-5}}$ | 0.0410 | 0.0105 |
| TimeMixer | 1.3054 | 0.2337 | 0.2251 | $9.30{\times}10^{-5}$ | **0.0430** | **0.0110** |
| iTransformer | **1.2902** | 0.3088 | 0.3019 | 0.0002 | 0.0365 | 0.0095 |

0.0110. All evaluated forecasts therefore retain only a small part of the target's local movement and residual fluctuation energy.

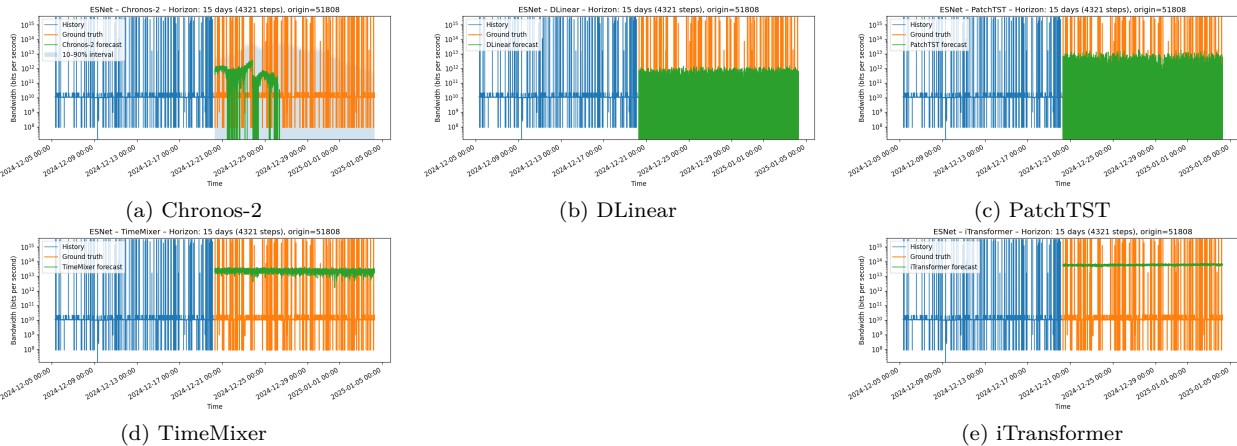

Figure 4: Forecasts on ESNet telemetry for a 15-day horizon, corresponding to 4321 time steps. The target contains abrupt changes over several orders of magnitude. None of the evaluated models reproduces these changes reliably.

The rolling-origin ESNet results are consistent with the plotted horizon. Different metrics select different models, but even the best average TVR and HFER values are only 0.0387 and 0.0099, showing that the forecasts contain only a small fraction of the target's local variation.

## 4.5 Rolling-origin variability

Table 6 reports the rolling-origin mean and standard deviation for the main settings discussed above. These are the values we use for model-level comparisons. The single-horizon tables correspond to the specific forecast segments shown in Figures 1–4; they explain those examples, but they are not used to rank models overall. When a plotted horizon and the rolling-origin average favor different models, we give more weight to the rolling-origin result because it averages over multiple non-overlapping test windows.

This distinction matters for ETTh1 at prediction length 96. On the plotted horizon, TimesFM has the lowest MSE, MAE, and DTW, while PatchTST has the lowest TDI. Across the six rolling origins, however, iTransformer has the lowest average MSE, TimesFM has the lowest average MAE and DTW, and PatchTST has the lowest average TDI. The plotted horizon is therefore useful for showing one concrete forecast, but the rolling-origin table gives the more stable summary of model behavior. We include the standard deviations because several metrics vary substantially across test windows; when the standard deviation is large relative to the mean, small differences in the mean should be interpreted cautiously. These standard deviations describe variation across deterministic forecast origins, not stochastic model uncertainty or confidence intervals.

The rolling-origin results also show why we avoid assigning a single overall winner. The model favored by MSE or MAE is not always the model favored by DTW or TDI, and neither necessarily preserves the most local variation. On ETTh1 at horizon 96, TimesFM has the lowest average MAE and DTW, while DLinear is closer to one under TVR and HFER. On ETTm1 at horizon 336, Chronos-2 has the lowest average pointwise errors, but DLinear preserves more local variation. At horizon 720, DLinear has the lowest average MSE and MAE, while iTransformer has the lowest average DTW. The ESNet rows show a different pattern: several models are close under pointwise scores, but all TVR and HFER values remain far below one. This indicates that the forecasts retain only a small fraction of the target's local movement, even when some pointwise or alignment scores look competitive.

## 5 Discussion and Limitations

The results suggest a practical way to read forecasting metrics. MSE and MAE are still useful summaries of pointwise error. DTW and TDI add information about alignment and timing. TVR and HFER add a

Table 6: Rolling-origin mean and standard deviation for the main settings. C2 = Chronos-2, TFM = TimesFM, DL = DLinear, PT = PatchTST, TM = TimeMixer, and IT = iTransformer. For MSE, MAE, DTW, and TDI, bold marks the lowest mean. For TVR and HFER, bold marks the mean closest to one. ETTh1 and ETTm1 use six rolling origins; ESNet uses three. TimesFM is omitted for ESNet because the 4321-step horizon exceeds the TimesFM horizon limit used in our experiments.

| Setting | Metric | C2 | TFM | DL | PT | TM | IT |
|---|---|---|---|---|---|---|---|
| ETTh1 / 96 | MSE | .103±.069 | .101±.073 | .135±.099 | .121±.061 | .138±.103 | **.098±.075** |
| | MAE | .259±.116 | **.252±.123** | .300±.141 | .289±.100 | .305±.138 | .253±.119 |
| | DTW | .211±.136 | **.174±.142** | .228±.171 | .218±.124 | .238±.169 | .201±.134 |
| | TDI | .016±.021 | .025±.038 | .013±.017 | **.008±.010** | .014±.021 | .014±.022 |
| | TVR | .283±.098 | .281±.123 | **.822±.139** | .522±.087 | .492±.089 | .415±.046 |
| | HFER | .377±.119 | .384±.194 | **.625±.114** | .497±.107 | .437±.134 | .372±.072 |
| ETTm1 / 336 | MSE | **.156±.157** | .163±.137 | .164±.162 | .160±.146 | .191±.168 | .170±.152 |
| | MAE | **.307±.197** | .325±.182 | .315±.205 | .320±.182 | .351±.197 | .329±.191 |
| | DTW | .228±.217 | **.212±.130** | .252±.226 | .252±.196 | .282±.229 | .260±.218 |
| | TDI | .032±.042 | .051±.055 | .010±.016 | .009±.010 | **.008±.010** | .011±.015 |
| | TVR | .120±.047 | .085±.020 | **1.128±.253** | .466±.048 | .560±.131 | .348±.079 |
| | HFER | .211±.142 | .174±.071 | **.502±.133** | .335±.098 | .352±.106 | .273±.108 |
| ETTm1 / 720 | MSE | .145±.170 | .179±.164 | **.134±.117** | .139±.138 | .158±.134 | .134±.110 |
| | MAE | .289±.190 | .331±.198 | **.283±.156** | .288±.168 | .312±.164 | .287±.147 |
| | DTW | .217±.194 | .253±.198 | .197±.142 | .199±.149 | .209±.146 | **.168±.098** |
| | TDI | .017±.019 | .009±.012 | **.007±.007** | .020±.028 | .016±.012 | .043±.054 |
| | TVR | .146±.040 | .058±.038 | **1.290±.254** | .686±.122 | .586±.143 | .341±.060 |
| | HFER | .194±.165 | .146±.141 | **.494±.055** | .446±.081 | .441±.045 | .471±.201 |
| ESNet / 4321 | MSE | 1.479±.147 | – | 1.478±.146 | 1.479±.143 | 1.467±.151 | **1.457±.162** |
| | MAE | .207±.018 | – | **.204±.017** | .214±.015 | .231±.005 | .279±.046 |
| | DTW | .175±.044 | – | **.152±.015** | .189±.007 | .219±.006 | .272±.049 |
| | TDI | .0151±.0251 | – | .00246±.00044 | .00025±.00027 | **.00005±.00004** | .00006±.00010 |
| | TVR | .001±.000 | – | .003±.001 | **.039±.018** | .031±.013 | .032±.016 |
| | HFER | .001±.000 | – | .001±.000 | **.010±.004** | .008±.003 | .008±.004 |

different check: whether the forecast has retained the local movement present in the target. These ratios should be read cautiously. A forecast with TVR or HFER close to one may still be wrong in level or phase, and a low pointwise error may still correspond to a visibly smoothed trajectory.

The study is limited in scope due to computing resource constraints. All experiments are univariate and target-only, so models that are designed to use cross-variable structure, especially iTransformer, are evaluated under a restricted setting. The reported MSE and MAE values are also target-column scores, not the multivariate aggregate scores usually reported for these datasets. Finally, the standard deviations are taken over deterministic rolling forecast origins. They describe sensitivity to the chosen test window, not stochastic model uncertainty.

The ESNet case also clarifies where TVR and HFER are useful. The target contains intermittent high-bandwidth bursts separated by long periods of relative inactivity. In this setting, pointwise and alignment-based metrics can make forecasts appear competitive even when the spike-like structure is largely missed. TVR and HFER expose this failure more directly because they measure how much local movement and short-scale residual energy remains in the forecast. However, these ratios are not a complete solution. ESNet-like telemetry also has features of anomaly or event prediction, where the main question is whether rare high-activity intervals are detected at the right time. Such an evaluation would require spike- or threshold-based metrics, but the dataset does not provide labeled anomaly intervals.

# 6 Conclusion

This paper examined cases where the usual pointwise metrics give an incomplete picture of forecast behavior. Across the benchmark and ESNet examples, low MSE or MAE does not always coincide with preservation of local movement. DTW and TDI help describe alignment and timing, while TVR and HFER make smoothing more explicit by comparing local variation and residual fluctuation energy.

The main lesson is not that one metric should replace another. Each metric answers a narrower question. Reporting pointwise error, temporal alignment, and local-variation ratios together gives a clearer account of how a forecast behaves, especially when peaks, abrupt changes, or short-term fluctuations are important for the downstream use case.

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

# A  Additional Results

## A.1  Omitted Benchmark Results

The main text plots four settings: ETTh1 at horizon 96, ETTm1 at horizons 336 and 720, and ESNet at the 15-day horizon. We chose these settings before inspecting model winners because they cover a short benchmark horizon, longer benchmark horizons, and a heavy-tailed telemetry case. The same evaluation code was run on the remaining benchmark settings listed in Table 7.

Table 7: Benchmark settings evaluated but not plotted in the main text. All use the same univariate target-only setup, scaling procedure, rolling-origin protocol, and metric definitions as the main results.

| Dataset | Prediction lengths omitted from main figures | Reason for appendix placement |
|---|---|---|
| ETTh1 | 336, 720 | Main text already includes ETTh1 at the short horizon. |
| ETTm1 | 96 | Main text already includes ETTm1 at medium and long horizons. |
| ETTm2 | 96, 336, 720 | Same benchmark family as ETTm1; included in the full sweep. |
| Weather | 96, 336, 720 | Included in the full sweep and used in the lookback check. |

The omitted settings support the same methodological point as the main text: metric choice changes which forecast property is emphasized. The ordering of models varies across datasets and horizons, so we avoid presenting a single model ranking. The released results include the rolling-origin outputs for these additional settings, which allows the plotted cases to be checked against the rest of the sweep.

## A.2  Lookback Sensitivity

We also ran a targeted lookback check for prediction length 720. The experiment compares lookback lengths of 336 and 720 for DLinear, PatchTST, and iTransformer on ETTm1 and Weather. The goal is to see whether the metric disagreements are mainly an artifact of using a 336-step input window for a 720-step forecast.

The effect of the longer lookback is model- and dataset-dependent. On ETTm1, PatchTST improves when the lookback increases from 336 to 720: its MSE drops from 0.0625 to 0.0464, and its DTW drops from 0.0985 to 0.0642. DLinear and iTransformer have lower MSE with the 336-step lookback. On Weather, the 336-step lookback gives lower MSE for all three models, although the 720-step lookback improves TDI.

These results do not remove the main finding. Changing the input length can improve particular models and metrics, but pointwise errors, alignment metrics, and local-variation ratios can still lead to different readings of the same forecast.

