# OpenReview forum: "Local Variation Matters: A Diagnostic Evaluation of Time Series Forecast Metrics"
_TMLR — Under review for TMLR_

### Review · Reviewer_Trbr · 2026-06-17

**Summary Of Contributions:**

# Summary
The work argues that two point-wise metrics that are widely used in evaluating long-horizon time-series forecasting tasks, MAE and MSE, have the problem of failing to capture some major failures in the prediction, like an abrupt peak or a major shift. The work studies two non-standard metrics for time series models on long-term forecasting, DTW and TDI, which measure structural similarity between the forecast series and the ground truth. It argues that they are complementary and diagnostic metrics. The work 'evaluates' the four metrics using mainstream time-series forecasting models on the ETT, weather, and ESnet datasets. Visual inspection on a few forecasting samples shows that DTW is indicative of major changes in the time-series. In the stress test, the work also shows

# Strength
The work is well-motivated based on good observations of time-series forecasting results and metric values.

# Weakness
The work describes empirical observations on a limited set of examples but stops short of (a) formulating the underlying problem in a way that admits scientific study, and (b) evaluating the four metrics in a systematic, comprehensive, and quantitative manner. There’s a weak connection between the quantitative study showing weak correlation between the four metrics' rankings and the claimed contribution. The rest of the experiment is essentially whether a metric number happens to agree with how a forecast looks based on the authors' eyeballing. That is a reasonable source of intuition, but it is not evidence that the metrics are correct, only that they are consistent with one reader's visual impression on a few cases. Please refer to my explanation below.

**Audience:**

No

**Audience Explanation:**

The general field of time-series models and the topic of evaluation protocols for long-term forecasting have a larger audience in the machine learning research community. Without seeing a proper statement of what problem the work is studying, I cannot assess whether there is an interested audience for this work or not.

**Broader Impact Concerns:**

Without seeing a proper statement of what problem the work is studying, it's pre-mature to assess its potential impact.

**Claims And Evidence:**

No

**Claims Explanation:**

1. The paper never operationally defines the phenomenon it cares about. The failures it attributes to MSE/MAE — smoothing of peaks, missed troughs, phase shifts, suppression of bursts — are described qualitatively and identified by eye, but never formalized into measurable quantities. Without a definition of the target property, there is no well-posed question against which any metric, including the two proposed ones, can be judged right or wrong. In particular, the notion of "visual faithfulness" that drives the entire argument is subjective and left undefined.
2. One of the major claims that DTW and TDI are complementary to MAE and MSE and useful diagnostic metrics is hand-wavy and barely contestable in this work. The author never materialized what a diagnostic metric actually means. To make things worse, the work claims that TDI "does not penalize" amplitude damping and it is an incomplete and sometimes misleading diagnostic.
3. The evaluation based on visual impression is anecdotal but not systematic or comprehensive based on hand-picked examples.

**Requested Changes:**

My opinion is that this work stands as a motivating observation instead of a rigorous scientific study. To bridge this gap, the authors could approach this challenge from the following perspectives:

1. Reframe the current study as motivation, and state a formal problem to study, e.g., DTW and TDI catch forecasting failures when phase changes happen and MSE/MAE miss these failures.
2. Propose a comprehensive and systematic framework to evaluate different metrics and ground your conclusions to the problem based on rigorous quantitative results, e.g. models that make more mistakes on time-series forecasting at the point of phase changes are better exposed by DTW and TDI metrics than MSE/MAE metrics.
3. I highly suggest the authors consider how to ground the notion of visual faithfulness in quantitative criterion as a starting point.
4. The work could include some recent time-series works into its evaluations, especially time-series foundation models (e.g. TimesFM, Moirai, TiRex) to attract more potential audience.

The problems and evaluation I suggested in the requested changes are just examples. I would leave it to the authors to frame the problem they intend to study and how they want to study it in a scientific and rigorous way.

---

> ### Author Response · Authors · 2026-06-29
>
> We thank the reviewer for the comprehensive critique. We agree that the original paper relied too much on visual inspection. The revision reframes the study around a measurable failure mode: loss of local variation through over-smoothing.
>
> ```text
> Limited examples; no well-formulated problem or systematic quantitative evaluation.
> ```
>
> We now define the target property as local-variation preservation and evaluate it with TVR and HFER alongside MSE, MAE, FastDTW, and TDI.
>
> ```text
> Weak connection between rank disagreement and the claimed contribution.
> ```
>
> We no longer make rank disagreement the main contribution; the revised argument is about pointwise error, temporal alignment, and local variation as distinct forecast properties.
>
> ```text
> The experiment relies on whether metrics agree with visual inspection.
> ```
>
> The figures remain for interpretation, but the claims are now grounded in explicit metric definitions and rolling-origin quantitative results.
>
> ```text
> The paper never operationally defines the phenomenon it cares about.
> ```
>
> We now define it as loss of local variation through over-smoothing, measured by relative total variation and residual fluctuation energy.
>
> ```text
> Smoothing, missed troughs, phase shifts, and bursts are described qualitatively.
> ```
>
> We formalize the smoothing component with TVR and HFER, while TDI measures temporal displacement and FastDTW measures alignment cost.
>
> ```text
> Without a target property, there is no well-posed metric question.
> ```
>
> The target property is now whether the forecast retains comparable local movement and short-scale residual energy to the target.
>
> ```text
> “Visual faithfulness” is subjective and undefined.
> ```
>
> We removed this framing; plots are examples, while the argument is based on the six reported metrics.
>
> ```text
> The claim that DTW/TDI are complementary diagnostics is hand-wavy.
> ```
>
> We narrowed this claim and now acknowledge CONTIME as prior work on DTW/TDI, while focusing our contribution on local-variation loss.
>
> ```text
> The paper does not define what a diagnostic metric means.
> ```
>
> We use “diagnostic” only to mean that a metric describes one forecast property and is not sufficient by itself.
>
> ```text
> The claim about TDI and amplitude damping is misleading.
> ```
>
> We revised the language: TDI measures displacement of the alignment path, not preservation of local amplitude variation.
>
> ```text
> The evaluation appears anecdotal and hand-picked.
> ```
>
> We added rolling-origin mean and standard deviation values and an appendix documenting benchmark settings not plotted in the main text.
>
> ```text
> Reframe the study around a formal problem.
> ```
>
> We formalize a related problem: whether forecasts preserve local variation, which better matches the observed smoothing behavior.
>
> ```text
> Propose a systematic framework for evaluating metrics.
> ```
>
> The revised framework reports pointwise, alignment, and local-variation metrics on the same forecasts.
>
> ```text
> Ground visual faithfulness in a quantitative criterion.
> ```
>
> We ground this through TVR and HFER rather than using visual faithfulness as an undefined criterion.
>
> ```text
> Include recent time-series foundation models.
> ```
>
> We added TimesFM, so the revised model set includes DLinear, PatchTST, TimeMixer, iTransformer, Chronos-2, and TimesFM.
>
> ```text
> Without a clear problem statement, the audience is unclear.
> ```
>
> The revised problem is when pointwise and alignment metrics fail to reveal loss of local variation, which is relevant when local fluctuations or burst-like behavior matter.

---

### Review · Reviewer_7qX9 · 2026-06-22

**Summary Of Contributions:**

**Summary**

This paper investigates whether Dynamic Time Warping (DTW) and the Temporal Distortion Index (TDI) provide complementary information to standard pointwise forecasting metrics such as MSE and MAE. The authors evaluate five forecasting models (DLinear, PatchTST, TimeMixer, iTransformer, and Chronos-2) on four benchmark datasets (ETTh1, ETTm1, ETTm2, and Weather) as well as ESNet network telemetry data, across multiple prediction horizons. Through quantitative results and forecast visualizations, they analyze the extent to which DTW and TDI capture temporal misalignments and shape distortions that may not be reflected by MSE and MAE. The study also examines the behavior of these metrics on heavy-tailed telemetry data characterized by abrupt spikes. The work positions DTW and TDI as complementary evaluation tools rather than alternatives to conventional error metrics.

**Strengths**

1. The paper addresses an important evaluation question in time series forecasting. Understanding the extent to which standard pointwise metrics capture temporal characteristics of forecasts is relevant for applications where the timing and shape of events matter.
2. The manuscript is clearly structured and easy to follow. The metrics, experimental setup, and empirical observations are presented in a coherent manner, and the inclusion of visual examples helps illustrate the reported results.

**Weaknesses**

1. The main finding (that DTW and TDI provide information that is not fully captured by MSE and MAE) largely confirms observations already reported in prior work, including the CONTIME paper cited by the authors. The paper does not introduce a new methodology, metric, or theoretical analysis beyond this empirical confirmation.
2. The experimental design raises questions regarding the choice of forecasting settings. A fixed lookback window of 336 timesteps is used for horizons of 96, 336, and 720 steps, including settings where the forecast horizon is equal to or substantially larger than the input context. The impact of this design choice on the observed forecasting behavior is not investigated.
3. The evaluation remains relatively narrow. Results are reported for a single target variable and a limited set of forecasting configurations, and the reported error values are not directly comparable to standard benchmark results. Additional analysis of key experimental choices would strengthen the study.
4. The ESNet experiments highlight limitations of all considered metrics on heavy-tailed telemetry data characterized by large spikes. While this observation is interesting, the discussion remains largely descriptive and does not explore possible directions for addressing this limitation.

**Audience:**

Yes

**Audience Explanation:**

Yes. The paper addresses a relevant question in time series forecasting evaluation, namely whether commonly used pointwise metrics adequately capture temporal characteristics of forecasts. Researchers and practitioners interested in forecast evaluation may find the empirical analysis and examples informative, even if the main conclusions are not entirely novel.

**Claims And Evidence:**

No

**Claims Explanation:**

While the empirical results generally support the specific observations reported in the paper, the evidence is not sufficient to fully substantiate the broader conclusions. In particular, the experimental design relies on a fixed lookback-to-horizon configuration whose influence is not analyzed, making it difficult to disentangle metric behavior from forecasting difficulty. In addition, the evaluation scope is relatively limited.

**Requested Changes:**

Please see wkeanesses

---

> ### Author Response · Authors · 2026-06-29
>
> We thank the reviewer for their valuable feedback. We address their concerns as follows in our updated manuscript.
>
> ```The main finding (that DTW and TDI provide information that is not fully captured by MSE and MAE) largely confirms observations already reported in prior work, including the CONTIME paper cited by the authors. The paper does not introduce a new methodology, metric, or theoretical analysis beyond this empirical confirmation.```
>
> As mentioned in our reply to reviewer Gwk3, we have added two new metrics Total Variation Ratio(TVR) and High Frequency Energy Ratio(HFER) that capture smoothing of local variations in a time series forecast.
>
> ```The experimental design raises questions regarding the choice of forecasting settings.```
>
> We have added a Lookback Sensitivity section where we test with lookback window 720 on the benchmark datasets for prediction length 720. While it does improve a lot of metrics, our point that pointwise forecast measures should not be interpreted as an automatic measure of superior performance among state of the art forecasting models holds true.
>
> ```The evaluation remains relatively narrow. Results are reported for a single target variable and a limited set of forecasting configurations, and the reported error values are not directly comparable to standard benchmark results. Additional analysis of key experimental choices would strengthen the study.```
>
> The error values are not directly comparable to benchmark results reported in the respective papers because many papers adopt different settings for lookback window, univariate/multivariate input, and so on. We chose fixed settings across different models to make them comparable within the scope of our experiments. Due to computing resource constraints, we cannot test all configurations, as this would also mean that we need to test them across the 6 different rolling windows for uncertainty testing.
>
> ```The ESNet experiments highlight limitations of all considered metrics on heavy-tailed telemetry data characterized by large spikes. While this observation is interesting, the discussion remains largely descriptive and does not explore possible directions for addressing this limitation.```
>
> Please refer to our reply to reviewer Gwk3 where we address this concern.

---

### Review · Reviewer_Gwk3 · 2026-06-23

**Summary Of Contributions:**

The paper distinguishes pointwise forecasting metrics (MSE, MAE) from temporal-alignment metrics (DTW, TDI), clarifying what aspect of forecast quality each one captures. Moreover, the empirical comparisons make a reasonably convincing case that MSE and MAE alone may fail to fully describe forecast quality, and that they need to be considered jointly with metrics that capture the shape and timing of a forecast (e.g. DTW, TDI). Finally, the authors discuss the limitations of DTW and TDI, showing through the ESNet case study that these alignment-aware metrics are not sufficient for highly intermittent, heavy-tailed time series.

**Strengths:**
- The motivation for the paper is clear and well-grounded, connecting the problem to real-world settings such as neutrino detection and financial volatility forecasting
- The paper evaluates five forecasting models with different architectures across four benchmarks and a real-world network telemetry dataset
- The figures of the forecasted univariate time series help reveal differences that the pointwise metrics alone do not capture

**Weaknesses:**
- The contribution of this paper relative to CONTIME is not clearly explained, especially since CONTIME already evaluates forecasting models using a combination of MSE, DTW, and TDI
- The paper does not provide any description (series length, number of variables, regularly/irregularly sampled) of the ETT and Weather datasets used in the experiments, beyond a citation
- The experimental setup and methodology are not fully documented in places. This is discussed more in the next text box
- The paper refers to DLinear as a state-of-the-art model, although it is now more commonly treated as a baseline in the literature

**Audience:**

Yes

**Audience Explanation:**

This is a well-written paper whose value is more methodological than empirically conclusive. It works better as a 'call' to the community to report a broader set of metrics when the timing and shape of the forecast matter. Its motivation is also grounded in a real-world problem, the NOvA experiment and related network telemetry, which broadens its relevance beyond standard ML benchmarking audiences to those working with operational or scientific forecasting data. The fact that this paper connects to prior work (CONTIME), which already uses a similar combination of metrics, with the exception of MAE, also suggests that there is already an interested audience for this topic

**Broader Impact Concerns:**

No broader impact concerns were identified

**Claims And Evidence:**

No

**Claims Explanation:**

The paper's central idea — that pointwise metrics can give an incomplete picture of forecast quality, and that DTW/TDI are useful complementary diagnostics — is a genuinely good and well-motivated one. However, a number of methodological gaps weaken confidence in the evidence:

- It is not clear which variable is being forecast in each dataset. For the ETT datasets, this can be inferred by consulting the dataset's own documentation, which specifies oil temperature (OT) as the designated target column; the paper itself does not state this. For the Weather it remains unclear which variable was used as the forecasting target. This makes it difficult to assess whether the choice of variable could affect the reported results, and limits reproducibility. For the ESNet dataset, the forecasting target appears to be bandwidth measured in bits per second, but this is only inferable from a plot rather than explicitly stated in the main text or methodology section
- In the rolling-origin evaluation, the reported standard deviation for TDI is comparable to or larger than the mean in most cases (e.g. DLinear's TDI of 0.0221 ± 0.0229 in Table 4). This raises questions about how reliable claims such as "DLinear again obtains the lowest average MSE, MAE, DTW, and TDI" actually are, and whether five rolling windows (three for ESNet) are a sufficient number to draw such conclusions, especially since no statistical significance testing is performed
- The paper states that results are reported only for "selected datasets of interest" at each prediction length, without explaining the criteria used to select them. This raises the question of whether the reported examples are representative or were chosen because they best illustrate the paper's argument
- The NOvA experiment in the introduction motivates why the timing and shape of local events matter in general, and ESNet is chosen as a practical stand-in for this kind of operationally important behavior. However, the ESNet results (Section 4.4) show that DTW and TDI do not reflect the true picture of model performance, since models that visibly fail to capture the extreme spikes can still receive low DTW and TDI scores. While the paper is transparent about this limitation, it weakens the claim that "they are useful when the timing and shape of peaks, troughs, bursts, and abrupt transitions matter, …"
- Although the paper reports both single-horizon and rolling-origin results, it does not provide a clear criterion for which of the two should be trusted when they disagree. For example, in the ETTh1 prediction-length-96 case, Chronos-2 performs best under MSE/MAE/DTW on the plotted horizon, while iTransformer performs best under the same metrics in the rolling-origin evaluation, and the paper does not indicate which result should carry more weight
- The paper does not clarify whether the iTransformer was given univariate or multivariate input. This matters because iTransformer's architecture relies on attention across variables — with only the target variable as input, this mechanism would have nothing to attend to, and the model would lose its main architectural advantage
- The ESNet experiments report results for only four of the five evaluated models — iTransformer is missing, with no explanation given anywhere in the paper for its absence

**Requested Changes:**

It is critical to clarify the paper's contribution relative to CONTIME, which already evaluates forecasting models using MSE, DTW, and TDI together. It is also important to report specific details that would make the work reproducible, such as the input variables provided to each model, the forecasting target column for each dataset, the size of the test region, etc., rather than requiring the reader to infer such details from the plots. In addition, the paper should explain why ETTh1 and ETTm1 were chosen for detailed discussion, rather than the Weather and ETTm2 datasets, despite both being introduced as part of the experimental setup.

---

> ### Author Response · Authors · 2026-06-29
>
> We thank the reviewer for taking the time to comprehensively review our paper. We address the reviewers' concerns as follows.
>
> ```The contribution of this paper relative to CONTIME is not clearly explained, especially since CONTIME already evaluates forecasting models using a combination of MSE, DTW, and TDI.```
>
> To address this concern, we introduce two metrics, Total Variation Ratio(TVR) and High Frequency Energy Ratio(HFER), which capture the smoothing of local variations that we described only through visual inspection earlier clearly.
>
> ```It is not clear which variable is being forecast in each dataset.```
>
> We have added these details in the Experiment Setup section now. For the weather dataset, we use a publicly available benchmark dataset[1] that already has a target column called OT. We do not make unsubstantiated claims about the exact target column name here since we are unsure what specific column this represents from the actual data from the German weather station.
>
> ```and whether five rolling windows (three for ESNet) are a sufficient number to draw such conclusions```
>
> This is due to computing resource constraints. The experiments take ~15 hours to run on a T4 GPU right now. Increasing the number of rolling windows any further is not feasible for this reason. We did however increase the number of rolling windows to 6 in the revised version of the paper, which is not a big change, but something worth being mentioned.
>
> ```This raises the question of whether the reported examples are representative or were chosen because they best illustrate the paper's argument```
>
> Our full results are available in the repository[2] linked in the paper. We have also added a small section in the appendix talking about the other results.
>
> ```While the paper is transparent about this limitation, it weakens the claim that "they are useful when the timing and shape of peaks, troughs, bursts, and abrupt transitions matter, …"```
>
> The new TVR and HFER metrics are more useful with the ESNet data, even though there is some scope for improvement with these metrics as well. This is because the ESNet data is characterized by intense intermittent spikes in network bandwidth followed by periods of relative inactivity that would not be accurately captured by any standard time series forecasting metric. Handling the ESNet use case as an anomaly prediction problem might work better, but there are no clear labelled anomalies in the dataset so it would be challenging to compute accuracy.
>
> ```Although the paper reports both single-horizon and rolling-origin results, it does not provide a clear criterion...```
>
> We have added details about this in the rolling origin variability subsection of the results.
>
> ```The paper does not clarify whether the iTransformer was given univariate or multivariate input.```
>
> iTransformer was given univariate input. We understand that iTransformer works best with multivariate input, but in the interest of time and uniformity among models, iTransformer was given univariate input.
>
> ```The ESNet experiments report results for only four of the five evaluated models — iTransformer is missing, with no explanation given anywhere in the paper for its absence```
>
> 5 out of 6 evaluated models are reported for ESNet now, with TimesFM being omitted from the list only for ESNet because ESNet's horizon length of 4321 is too large for the horizon length of 1024 expected by TimesFM without finetuning.
>
>
> [1] Kwangryeol Park. Time Series Forecasting Datasets. Hugging Face dataset repository, 2024. URL https:
> //huggingface.co/datasets/pkr7098/time-series-forecasting-datasets. Accessed: 2026-06-29
>
> [2] https://anonymous.4open.science/r/Rethink_time_series_forecasting_metrics-262E